# A determinate method of metrology attribute benchmark of commercial banks' management efficiency

**Ren Jing[1], Xin Chang[2]***

1 School of Management Engineering, Xi'an University of Finance and Economics, Xi'an, China, 2 School of Electronic Information Engineering, Xi'an Technological University, Xi'an, China

* cx3020784292@163.com

**Data Availability Statement:** All relevant data are within the article and its Supporting information files.

**Funding:** Soft science research plan project in Shaanxi Province (2016KRM018) REN Jing.

## Abstract

The evaluation or metrology of commercial bank management efficiency is the core of its effective management. The existing management efficiency of commercial banks adopts evaluation scheme rather than metrology. There are four shortcomings with the evaluation scheme, the evaluation is a ranking of advantages and/or disadvantages, all objects should participate in the evaluation, the evaluation results are valid only for the evaluated objects, and the evaluation results lack the metrology benchmark of the object domain. To address this issue, the paper presents a method to determine the benchmark of management efficiency, which is similar to the unit "meter" in the metrology of "length" and the unit "scale" in the metrology of "management efficiency". Firstly, the method of cluster analysis is used to solve the problem of attribute metrology base of management efficiency metrology. Based on the master of certain attribute data of commercial bank management efficiency metrology, cluster analysis is carried out onto the attribute data, and its characteristics and patterns are mined to determine the candidate benchmark set. Secondly, in the candidate datum set, k-means method is used to determine the metrology attribute datum to obtain the general metrology attributes. Finally, the absolute metrology of management efficiency is carried out for any commercial bank according to the benchmark, and the validity and feasibility of the benchmark are verified with an example. In such a way, the deficiencies of four aspects of evaluation are solved. Such a strategy can be adapted to different banks at any different time for their respective measurement, which extends the clustering statistical methods for attribute datum determination. The results can be applied to some other fields wherein object metrology is the basic task.

## 1 Introduction

Since the start of 21st century, the domestic and international economic circumstances are intricate, and the strife in the financial business is progressively furious. In recent years, commercial banks are assuming an inexorably significant position in banking world and playing a pivotal role in financial business. While the management efficiency is the core of bank operation and management, also are both the competitive advantages of banks and the key to

**Competing interests:** The authors have declared that no competing interests exist.

preventing financial risks [1]. The principle motivation behind administration is to improve efficiency, yield benefits and accomplish better administrative outcomes. Hence, management efficiency assumes the back bone of commercial banking system. It is only by improving their management efficiency that they can be full of vitality and remain invincible in fierce competition. Subsequently, one of the key issues facing management sciences is how to improve management productivity and give the metrology method of management efficiency. At present, research on the evaluation and measurement of the management efficiency of commercial banks focuses mainly on factors of influence and management efficiency measurement methods.

The literature in [2–4] analyzes two factors, external variables (macro factors and medium) and internal elements (micro factors), that influences the productivity and efficiency of Chinese commercial banks. Macroscopic factors primarily indicate to the macroeconomic level, medium factors chiefly incorporate the market structure, and the micro factors principally involves the structure of property rights, internal management, scale, capital management and capital, capital adequacy, organizational structure, intermediary business, and so forth.

The research on evaluation method mainly includes financial attribute analysis and frontier analysis methods. The former study was proposed by Alhadeff (1954), who focused the productivity of 210 banks in California from 1938 to 1950 [5]. Literature [6] assessed the liquidity, profitability and security of banks from the perspective of operating capacity. Study [7] broke down the information of 10 commercial banks of China in 2004, and reasoned that joint-stock commercial banks had higher characteristics in the financial attribute analysis model than the four state owned banks. Because of the convenience of data acquisition of financial attribute analysis method, it is widely use in recent days. Moving on, study [8] used stochastic frontier analysis method to evaluate the proficiency of 214 banks in the United States of America (USA) during 1991–1992 and found that the average efficiency of sample banks was low. In literature [9] productivity index was used to study the productivity changes of commercial Banks of different sizes in the USA. Literature [10] calculated the efficiency output of Spanish banks from 1993 to 1995 by using data envelope analysis method, and concluded that the size inefficiency was the main reason for the technical inefficiency of Spanish banks. Literature [11] utilized data envelopment analysis (DEA) model to estimate the efficiency of Turkish banking industry from 1988 to 1996, and the outcomes indicated that the low level of technical efficiency was the primary reason behind the trivial productivity of Turkish banking industry. Literature [12, 13] compares the efficiency of state-owned commercial and foreign commercial banks, and concludes that the average efficiency of foreign banks in developed countries is higher than that of state-owned banks, while the opposite conclusion is drawn in developing countries.

As per the existing literatures [14–16], the research on bank management efficiency focuses on evaluation, which is a comparative study rather than absolute metrology. There are following issues in evaluation: (i) Evaluation is a ranking of the pros and cons. (ii)All the evaluated objects should participate in the evaluation. (iii)Evaluation results are only valid for the objects participating in the evaluation. (iv) Evaluation results lack the object domain standard. (v) The addition of evaluation objects need to be re-evaluated. From a perspective on the above problems, be similar to the meter ruler in measuring length, could we consider whether the bank management efficient can also be metrology? As a consequence of the length with the worldwide arrangement of units "m" as a unit, regardless of where in the world, as long as this reference metric is used, not only do we know the metrology origin of all kinds of length ratio, but we can also understand the universal value of length, thereby solving the four significant problems of the evaluation. In this regard, literature [17–19] respectively discussed the metrology, weighing estimation, and quantification methods of enterprise management efficiency. This strategy is additionally reasonable for the investigation of the management efficiency attributes

in commercial banks. The index of the Management Efficiency Metrology Attributes in Commercial Banks (MEMACB) is not a metric for the quality of efficiency, but a benchmark for the importance of attributes. The attributes of the quantitative values of each variable should therefore be considered. But there is a new problem with metrology: the identification of benchmarks is just as critical as the determination of "meters".

To sum up, this paper introduces a clustering evaluation method to determine the benchmark for management efficient metrology attributes. Firstly, the connotation and characteristics of the metrology attribute benchmark are defined for bank management efficiency. Based on this, a cluster mining technology is used to analyze the metrology attribute data of management efficiency of several commercial banks through cluster analysis, and to min its characteristics and patterns, while determine its candidate benchmark set. Secondly, in the candidate datum set, k-means method is used to determine the metrology attribute datum and obtain a general metrology attribute datum (scale). Finally, the absolute metrology of management efficiency can be carried out for any commercial bank according to the benchmark, and the validity and feasibility of the benchmark are verified by an example. Such a scheme solves the problem of determining the benchmark of commercial bank management efficiency metrology.

## 2 Metrology attribute reference and description of proposed methodology

### 2.1. Benchmark analysis of MEMACB

**2.1.1 The benchmark connotation of MEMACB.**   Reference indicates to the standard utilized as the beginning reference in metrology work, and for the most part refers to the attribute reference. At the same time, benchmark is a broadly utilized concept in mechanical manufacturing, mechanical products from the design of the part size labeling, the positioning of the work-piece during manufacturing, dimension metrology during calibration, until the assembly of parts assembly location determination, all need to utilize the idea of benchmark.

Metrology: refers to "the process of verification with a specified base known quantity as a unit, compared with unknown quantities of the same type".

It is understood from definition of metrology that two components comprise: one is benchmark of metrology attributes and the other is metrology attributes (unknown quantities of the same type). Therefore, so as to quantify the proficiency of bank management, the problem of metrology attribute benchmark must be addressed. The metrology attribute benchmark in Bank management efficiency is the fundamental objective, primary concern or beginning stage dictated by bank management efficiency estimation.

**2.1.2 Benchmark characteristics of MEMACB.**   Compared with the benchmark of industrial metrology, the benchmark of **MEMACB** has the following characteristics:

Benchmark is a specified known quantity. When all objects are based on this known quantity. It is therefore not only necessary to calculate the management performance of commercial banks in the past and present, but also to estimate their potential profitability. If the choice of the metric is not sufficient, it will not only cause problems for the quantitative calculation of the importance of the management performance in commercial banks, but it may also not be suitable for the development of society, so it is necessary to have an acceptable benchmark assessment policy. The benchmark determination of **MEMACB** is a process. First of all, it is imperative to master the fundamental circumstance of management efficiency attributes in most commercial banks. The benchmark has nothing to do with the size of the commercial bank, but has something to do with the value of the management attribute in the commercial bank. After understanding the value of management attributes, it is necessary to determine the status of management attributes in most commercial banks. Second, an important task is to

determine the basic concept, definition and calculation formula of the benchmark attributes. The attributes determined should reflect the overall status of management efficiency attributes in most commercial banks. These attributes do lay the foundation for the development of effective, feasible and comprehensive metrology in commercial bank management.

The benchmark of **MEMACB** in commercial bank is established on the basis of commercial bank management efficiency statistical data. In order to establish a benchmark for the MEMACB, it is important to gather statistics and a description of the management efficiency of major commercial banks in recent years in order to recognize the improvement in management performance, predict future changes in the management efficiency of commercial banks and to take a comprehensive view of the different situations. Through the above study, it can be concluded that the benchmark determination of the **MEMACB** must take into account various influential factors and make comprehensive selection.

**2.1.3. Metrology attributes reference.** Attribute benchmark is the basic management efficiency target of commercial banks. The aim of management efficiency is to focus on the efficient functioning of commercial banks and improve profitability. Efficiency benchmark is the basic principle to judge the efficiency of commercial banks. In practice, the efficiency benchmark can be used to give the effectiveness of the management structure, organizational structure and operating mechanism of commercial banks, so as to promote their continuous innovation and further improvement.

## 2.2 Description of metrology attributes reference method

**2.2.1. Selection of metrology attribute reference determination method.** The selection and assessment of the benchmark may be made on the basis of the following two aspects: (i) The selection of the benchmark may be based on public opinion and rational judgement. (ii) In each attribute, the outstanding performance of **MEMACB** can be used as the benchmark, but in practice it is not feasible because, in order to obtain all the best commercial banks, truth cannot be identified.

Aiming at the shortcomings of the above two aspects, according to the attribute characteristics described above, cluster mining technology is used to mine the characteristics and patterns of **MEMACB** through cluster analysis, so as to determine their candidate benchmark set. On this basis, the k-means method is used to select the appropriate benchmark according to the established goal of metrology, so as to make value of the **MEMACB** more objective and accurate, so as to solve the benchmark selection problem of **MEMACB**. Only the method meeting the following two conditions can be considered as a benchmark method. First, the reference method must be specific to the substance defined; Second, all parameters, modified values depending on other substances or substrates, must be known or can be calculated with appropriate uncertainty. Therefore, the above method is not innate or benchmark method. Only when they have the above properties at the same time, is the real feasible benchmark method.

The specific selection process is as follows:

Step 1: collect representative values of **MEMACB**;

Step 2: adopt clustering method to classify the **MEMACB**;

Step 3: take the class containing the most concentrated metrology attributes of commercial Banks as the benchmark candidate set of this attribute;

Step 4: calculate the average value of the management metrology attribute value of each commercial bank in the candidate class as the attribute benchmark. The metrology attribute benchmark is determined.

### 2.2.2 The metrology attribute benchmark is determined.

1. Theory and data preparation

(i) Metrology attributes to be obtained through a lot of analysis data.

(ii) Data standardization.

In order to give the management efficiency of commercial banks, it is necessary to select metrology attributes that reflect the management characteristics of commercial banks in various aspects. However, the dimensions of each metrology attribute are not the same, and the order of magnitude of metrology attribute values is also quite different. Therefore, it is necessary to conduct standardized processing on the data, unify the dimensions of each metrology attribute, and convert the information with practical significance into standard data.

In this paper, the metrology of management efficiency in each commercial bank is taken as a data, that is, vector. The each attribute of the commercial bank management efficiency is the vector component. After the design quantity attribute value, the $m$ efficiency attribute value vector of commercial bank management is obtained and the matrix $X$ of $m{\times}n$ dimension is formed. Where, the number of commercial Banks is $m$, and the number of attributes is $n$. The matrix $X$ is normalized so that the average value of each variable is 0 and the variance is 1.

$$x' = x_{ij} - \frac{\bar{x}_j}{\sqrt{Var(x_j)}} \quad (i=1,2,..,m,j=1,2,\ldots,n) \tag{1}$$

Among them

$$\sqrt{Var(x_j)} = \frac{1}{m-1} \sum_{i=1}^{m} (x_{ij} - \bar{x}_j)^2, \quad \bar{x}_j = \frac{1}{m} \sum_{i=1}^{m} x_{ij}$$

$x_{ij}$ is the attribute value of the $j^{\text{th}}$ attribute of the $i^{\text{th}}$ metrology object, and $\bar{x}_j$ is the average value of the $j^{\text{th}}$ attribute.

The matrix $X'$ of $m \times n$ dimensions is thus formed, let's call it matrix $Y$ (where each vector represents the efficiency characteristic of a commercial bank attribute metrology).

2. Clustering mining implementation

The attributes of different commercial banks have different characteristics and their respective emphases. In order to select the appropriate benchmark in a targeted way, this study adopts k-means method to cluster the data, and on this basis analyzes and selects the benchmark.

i) Applicability of the method

K-mean method is a kind of clustering mining technology based on density. It is a kind of typical data mining technology. This method can divide the clustering boundary according to different requirements. In the process of selecting the aggregation point, the selection of the aggregation point is optimized dynamically by comparing the internal information. The division of clustering boundary is related to the parameters given in advance describing the spatial distance parameter $S(i \neq j)$ between category $i$ and category $j$ and the concentration parameter P describing the number of spatial ranges of category $i$. According to experience, by adjusting the class distance parameter $S$ and class density parameter $P$, the classification boundary can be determined according to the level of commercial bank attributes that need to be considered in practice, and the classification refinement degree can be controlled.

ii) Clustering step

To solve the benchmark selection problem, the "density" of each point is used to determine the aggregation point and the number of classes. The specific steps are as follows.

Step 1 is to find the "density" $p_i$ of each point.

Determine a positive number $s_0$, take each sample point as the center of $x(i)$ and $s_0$ as the radius to make a hypersphere in n-dimensional space; $s(x(i), x(j))$ is the distance from $x(i)$ to $x(j)$ between the two.

$$s(\boldsymbol{x(i)},\boldsymbol{x(j)})=\sum_{h=1}^{n}(\boldsymbol{x(i)}_h-\boldsymbol{x(j)}_h)^2 \tag{2}$$

If $s(\boldsymbol{x(i)}, \boldsymbol{x(j)}) < \boldsymbol{s_0}$, then $x(j)$ falls within the super sphere. The total number of points that fall within the super sphere is the density $p_i$ of the $x(i)$ point. It can be seen that the greater the $p_i$-density, the greater the qualification of $x(i)$ will be as a coagulation point.

Step 2 is to determine the condensation point.

According to the above-bound method, two parameters $S > 0$, $P \geq 0$ are set to determine the condensation point. The points with the largest and the second-largest density $p_i$ are taken as condensation points. Let the condensed point set belong to $E$. If there are already $k$ condensed points $e_i \in E$, $i = 1, 2, \ldots, k$, then for the $j$ point $x(j)$, consider the distance $s(e_i–x(j))$ and the density $p_j$ between the point and the known condensing points, where

$s(e_i - x(j)) = \sum_{h=1}^{n}(e_{ik} - x(j)_h)^2$. If $s(\boldsymbol{e_i–x(j)}) > S$, $i = 1, 2, \ldots, k$ is satisfied and the density of

this point $p_j > P$, then $x(j) \in E$, so it is the next condensation point. By repeating the process, a batch of condensation points can be selected. The number of condensation points is the number of classes to be gathered.

Step 3 is clustering.

When $K$ condensed points are selected, it is assumed that each type is $C(i)$, $i = 1, 2, \ldots, K$. For the remaining $N – K$ points, calculate the distance $s(e_i–x(j))$, ($i = 1, 2, \ldots, K$, $j = 1, 2 \ldots$, $N – K$, from them to each condensed point. If $s(e_i - x(j)) = \min_i(s(e_i - x(j)))$, then $x(j) \in C(k)$; the point is classified into the class represented by the nearest condensed point.

3. **Select the benchmark**

According to the characteristics of candidate benchmark, the candidate reference set is determined according to the clustering results, and the benchmark is selected on the basis of it.

i) Characteristics of candidate benchmark

The different condensation points (or the center of gravity) form a candidate reference array reflecting specific characteristics, such as the metrology points of certain forms of commercial banks. Through evaluating the data characteristics of the compressed point (or center of gravity) of the cluster, the properties of all types of commercial banks can be identified, the quality of the benchmark of the unknown capital of the established commercial bank can be finalized and the characteristics of each candidate benchmark can be evaluated. In addition, the class distance parameter $S$ and the class density parameter $P$ can be adjusted to divide the data into a finer class, which is determined by the comparison reference requirements. The adjustment of the parameters $P$ and $S$ in operation reflects the purpose of the classification and can be divided into small classes of the gross level or small class of the fine hierarchy, and the size classes are highly compatible. Classification and research can therefore be carried out conveniently in accordance with the requirements by adjusting the parameters.

ii) Determination of attribute Datum

For a certain attribute, the class with the maximum value corresponding to the attribute is considered to be the benchmark class for the attribute according to the candidate benchmark set and its characteristics as mentioned above, since it reflects the basic characteristics of the attribute. Then the average value of the management metrology attribute value of each commercial bank in this class is calculated as the attribute benchmark. So far, the determination of the metrology attribute benchmark of commercial Banks has been realized.

iii) Metrology object attribute calculation

The metrology object attribute is the value of the metrology object attribute relative to the defined attribute reference Likewise, the value of the metric attribute of the giving object can be determined by comparing the attribute of each object to be given with the reference.

## 3 The case study

### 3.1 Example calculation

An example is given to illustrate the process of determining the benchmark of this method. Considering the inaccessibility of data, taking commercial banks as an example, the ability to contribute to society, such as net interest income, net profit, non-performing loan ratio and net assets per share, is regarded as the metrology attribute of corporate governance efficiency of commercial banks [2, 20, 21]. In this example, the metrology attributes and attribute sample data are shown in Table 1. Table 1 has 30 groups of commercial bank attribute data, and each type of commercial bank management metrology attribute contains 6 attribute evaluation values.

The original data of attributes in Table 1 are preprocessed by formula (2). Then we utilize formula (1) and SPSS software to determine the candidate benchmark set, to lay a foundation for selecting a suitable benchmark, and then calculate the average value of the same attribute of each commercial bank in the candidate benchmark class as the reference value of this attribute. The processing results are shown in Tables 2–5.

Tables 2 and 3 show the final clustering center and the distance between the final clustering center.

Tables 4 and 5 respectively represent the class to which each observation belongs and the number of cases in each clustering center. According to the clustering algorithm, we can see that all the observations are divided into 6 categories according to the distance from the clustering center, and the number of bank cases in the second category is the largest.

To better reflect the characteristics of clustered classes and attributes, the average value of the attribute samples of the number of clustered cases in the same category was taken as the reference value.

First, select the attribute benchmark based on the data in Table 2, that is, the one with the largest value of the attribute in the 6 clustered classes is selected as the attribute benchmark. For example, the attribute value of "non-performing loan ratio" in the first category of the cluster of 6 categories is 0.8488 at most. The attribute value of "average return on assets" in category 3 of the 6 categories in the cluster is 0.9909. The maximum value of "net interest income" in category 6 of the 6 categories is 0.9143. The maximum attribute value of "net profit" in the 6th class in the aggregate 6 class is 0.9080; The maximum value of the attribute of "basic earnings per share" in category 6 is 0.9167. The maximum attribute value of "net asset per share" in category 2 of the 6 aggregated categories is 0.7474.

**Table 1. Sample data table of metrology attributes for management efficiency of commercial banks.**

| Output attribute | | | Non-performing loan ratio (%) | Average return on assets (%) | Net interest income (million) | Net profit (million) | Basic earnings per share | Net asset per share |
|---|---|---|---|---|---|---|---|---|
| Bank | No. | years | | | | | | |
| ICBC | 1 | 2010 | 1.08 | 1.32 | 303749 | 166025 | 0.48 | 2.35 |
| | 2 | 2011 | 0.94 | 1.44 | 362764 | 208445 | 0.60 | 2.74 |
| | 3 | 2012 | 0.85 | 1.45 | 417828 | 238691 | 0.68 | 3.22 |
| | 4 | 2013 | 0.94 | 1.44 | 443335 | 262965 | 0.75 | 3.63 |
| | 5 | 2014 | 1.13 | 1.40 | 493522 | 276286 | 0.78 | 4.23 |
| | 6 | 2015 | 1.50 | 1.30 | 507867 | 277720 | 0.77 | 4.80 |
| | 7 | 2016 | 1.62 | 1.20 | 471846 | 279106 | 0.77 | 5.29 |
| Bank of Communications | 8 | 2010 | 1.12 | 1.08 | 84995 | 39042 | 0.66 | 3.96 |
| | 9 | 2011 | 0.86 | 1.19 | 103493 | 50735 | 0.82 | 4.39 |
| | 10 | 2012 | 0.92 | 1.18 | 120126 | 58373 | 0.88 | 5.12 |
| | 11 | 2013 | 1.05 | 1.11 | 130658 | 62295 | 0.84 | 5.65 |
| | 12 | 2014 | 1.25 | 1.08 | 134776 | 65850 | 0.88 | 6.34 |
| | 13 | 2015 | 1.51 | 1 | 144172 | 66528 | 0.90 | 7.00 |
| | 14 | 2016 | 1.52 | 0.87 | 134871 | 67210 | 0.89 | 7.67 |
| Construction bank | 15 | 2009 | 1.15 | 1.24 | 211885 | 106756 | 0.45 | 2.39 |
| | 16 | 2010 | 1.14 | 1.32 | 251500 | 134844 | 0.56 | 2.80 |
| | 17 | 2011 | 1.09 | 1.47 | 304572 | 169258 | 0.68 | 3.27 |
| | 18 | 2012 | 0.99 | 1.47 | 353202 | 193602 | 0.77 | 3.80 |
| | 19 | 2013 | 0.99 | 1.47 | 389544 | 214657 | 0.86 | 4.30 |
| | 20 | 2014 | 1.19 | 1.42 | 437398 | 227830 | 0.91 | 5.01 |
| | 21 | 2015 | 1.58 | 1.30 | 457752 | 228145 | 0.91 | 5.78 |
| | 22 | 2016 | 1.52 | 1.18 | 417799 | 231460 | 0.92 | 6.28 |
| Agricultural Bank | 23 | 2009 | 2.91 | 0.82 | 181639 | 65002 | 0.25 | 1.32 |
| | 24 | 2010 | 2.03 | 0.99 | 242152 | 94907 | 0.33 | 1.67 |
| | 25 | 2011 | 1.55 | 1.11 | 307199 | 121956 | 0.38 | 2.00 |
| | 26 | 2012 | 1.33 | 1.16 | 341879 | 145131 | 0.45 | 2.31 |
| | 27 | 2013 | 1.22 | 1.20 | 376202 | 166211 | 0.51 | 2.60 |
| | 28 | 2014 | 1.54 | 1.18 | 429891 | 179461 | 0.55 | 3.05 |
| | 29 | 2015 | 2.39 | 1.07 | 436140 | 180582 | 0.55 | 3.48 |
| | 30 | 2016 | 2.37 | 0.99 | 398104 | 183941 | 0.55 | 3.81 |

Note: the data is collected from the annual report of each bank website.

**Table 2. Final aggregation center.**

| | Gather | | | | | |
|---|---|---|---|---|---|---|
| | 1 | 2 | 3 | 4 | 5 | 6 |
| Non-performing loan ratio | .8488 | .4040 | .3322 | .7216 | .4278 | .4891 |
| Average return on assets | .6156 | .7298 | .9909 | .7347 | .8333 | .8844 |
| Net interest income | .4172 | .2400 | .6862 | .8297 | .5882 | .9143 |
| Net profit | .2865 | .2099 | .7689 | .6497 | .5022 | .9080 |
| Basic earnings per share | .3152 | .9115 | .7862 | .599 | .5127 | .9167 |
| Net asset per share | .1949 | .7474 | .4555 | .449 | .3140 | .6821 |

**Table 3. Distance from the final aggregation center.**

| Gather | 1 | 2 | 3 | 4 | 5 | 6 |
|---|---|---|---|---|---|---|
| 1 | | .953 | 1.001 | .691 | .595 | 1.198 |
| 2 | .953 | | .828 | .911 | .752 | .989 |
| 3 | 1.001 | .828 | | .536 | .458 | .419 |
| 4 | .691 | .911 | .536 | | .449 | .553 |
| 5 | .595 | .752 | .458 | .449 | | .759 |
| 6 | 1.198 | .989 | .419 | .553 | .759 | |

Then, based on the above chosen attribute benchmark, the number of observed values in each cluster is present in Table 5, the case number corresponding to the cluster class is in Table 4, and the sample data corresponding to the serial number of metrology objects can be found in Table 1. The average value is calculated as the attribute benchmark value. Calculates the, such as: the non-performing loan ratio in the six classes of gathered in the first in the class

**Table 4. Group members gathered.**

| Case number | Bank | Gather | distance |
|---|---|---|---|
| 1 | ICBC | 5 | .127 |
| 2 | ICBC | 3 | .171 |
| 3 | ICBC | 3 | .177 |
| 4 | ICBC | 3 | .244 |
| 5 | ICBC | 6 | .216 |
| 6 | ICBC | 6 | .159 |
| 7 | ICBC | 6 | .156 |
| 8 | Bank of Communications | 2 | .319 |
| 9 | Bank of Communications | 2 | .226 |
| 10 | Bank of Communications | 2 | .146 |
| 11 | Bank of Communications | 2 | .056 |
| 12 | Bank of Communications | 2 | .101 |
| 13 | Bank of Communications | 2 | .224 |
| 14 | Bank of Communications | 2 | .319 |
| 15 | Construction bank | 5 | .213 |
| 16 | Construction bank | 5 | .162 |
| 17 | Construction bank | 3 | .197 |
| 18 | Construction bank | 3 | .100 |
| 19 | Construction bank | 3 | .200 |
| 20 | Construction bank | 6 | .174 |
| 21 | Construction bank | 6 | .147 |
| 22 | Construction bank | 6 | .219 |
| 23 | Agricultural Bank | 1 | .187 |
| 24 | Agricultural Bank | 1 | .187 |
| 25 | Agricultural Bank | 5 | .185 |
| 26 | Agricultural Bank | 5 | .105 |
| 27 | Agricultural Bank | 5 | .186 |
| 28 | Agricultural Bank | 4 | .211 |
| 29 | Agricultural Bank | 4 | .104 |
| 30 | Agricultural Bank | 4 | .130 |

**Table 5. Number of observations in each cluster.**

| Gather | | |
|---|---|---|
| Gather | 1 | 2.000 |
| | 2 | 7.000 |
| | 3 | 6.000 |
| | 4 | 3.000 |
| | 5 | 6.000 |
| | 6 | 6.000 |
| Effective | | 30.000 |
| Missing | | 0.000 |

attribute value up to 0.8488, as a result, the attribute datum gather in class 1, according to Table 5, number of observed value for class 1 is 2.000, according to the measuring object number 23, 24 in Table 1 the corresponding attribute value of the non-performing loan ratio were 2.91, 2.03, and its average value is 2.47, so it is concluded that the properties of the basic value of 2.47. Similarly, the base values of the other five attributes mentioned-above can be obtained as follows: 1.4567, 464364 (RMB¥ten thousand), 253424.5 (ten thousand yuan), 0.8433 and 5.7329.

## 3.2 Method validity analysis

The current management efficiency evaluation or metrology is generally based on the model $y_i = \sum_{i=1}^{n} w_i x_{ij}$ (where $w_i$ is the attribute weight, $x_{ij}$ is the attribute value, $y_i$ is the value of management efficiency), and the model is used to calculate and compare.

The method in this paper first calculates the benchmark value of the metrology attribute, then uses the weight determination method in the literature [15] to determine the attribute weight, and at last calculates the management efficiency value according to formula (3),

$$z = \sum_{i=1}^{n} \frac{x_i}{\bar{x}_i} \times w_i \qquad (3)$$

where $w_i$ is the attribute weight, $x_i$ is the attribute value, $\bar{x}_i$ is the attribute reference value and $z$ is the management efficiency value.

For example, the weights of 6 attributes in the sample data determined by the weight determination method in literature [15] were 0.02, 0.12, 0.29, 0.45, 0.08, 0.04, respectively. We calculated the value of ICBC's management efficiency in 2014, 2015 and 2016 by using the above management efficiency metrology formula to be 0.9665, 1.0268 and 1.0361. Similarly, the management efficiency of bank communications in 2014 can be calculated as 0.4279 respectively. The annual management efficiency of other banks can be calculated separately.

This method can be used to give the management efficiency value of any metrology object, thus solving the six shortcomings of management efficiency evaluation:

i) Give the specific management efficiency value instead of sorting the advantages and disadvantages. It is revealed from the proposed method that the determined management efficiency attribute benchmark is similar to the "meter stick" of length metrology. With this "meter stick", the management efficiency of any commercial bank can be given, and the given value has the feature of "consistency". Thus, the units of metrology of science and technology, production and daily life are unified into one unit. However, by using the evaluation method, only the good and bad ranking can be obtained.

ii) The efficiency value of a single metrology object can be given without all metrology objects participating in the metrology. As mentioned above, the benchmark "meter stick" can be used to give the management efficiency of any commercial bank in a given year, without the participation of other commercial banks in different years. Evaluation is different. All evaluation objects must participate in the evaluation. As long as the attribute value is changed, all evaluation objects must participate in the evaluation again.

iii) The value of metrology results is as standard as "length". Metrology in the same benchmark, obviously the benchmark is the standard, once the standard is determined, metrology to comply with this standard naturally has the significance of the qin shi huang unified metrology.

iv) Adapt to separate metrology in different time periods of different Banks. Due to the determination and unification of metrology benchmark, the management efficiency of different commercial Banks in different time periods can be given.

v) The proposed clustering statistical method has been extended and applied in the new field of attribute benchmark determination. There are few types of research on the metrology of management efficiency of commercial banks, and no relevant literature can be found when applying it to the determination of the benchmark of metrology. According to the validity analysis, this method is reasonable and effective in this field, so it expands the applications of this clustering statistical method.

## 3.3 Reasons for establishing a metrology attribute benchmark instead of a metrology benchmark

Efficiency evaluation method based on the particularity of the efficiency of management, management of diversity and the imperfect of the qualitative research, establish a benchmark for attribute has great flexibility, for follow-up study and determination of the metrology standard and units to provide a larger space, on the one hand, according to the in-depth study, can add new attributes, original benchmark properties do not need to change, on the other hand, can study more effectively and more reasonable weights of attributes, so that the efficiency of management metrology and research application more reasonable, more open. On the other hand, if the metrology benchmark is established, it does not have the above advantages.

## 4 Conclusion

Conclusions drawn by the article are as follows: (1) a new method to determine the benchmark of commercial bank management efficiency metrology is presented; (2) The obtained management efficiency metrology benchmark can be used as a scale like the "meter" in the metrology of "length", and such a scale can be used to measure any one of the measured objects individually, rather than all objects participating in the metrology, which greatly improves the metrology efficiency; (3) Absolute numerical value of management efficiency are measured for the object according to the metrology scale, rather than the "ranking of multiple objects in the evaluation"; (4) This method adapts to independent metrology of different banks in different periods. (5) The clustering statistical domain methods are extended in attribute benchmark determination.

## Supporting information

**S1 File.**
(DOCX)

## Author Contributions

**Data curation:** Ren Jing, Xin Chang.

**Formal analysis:** Ren Jing, Xin Chang.

**Writing – review & editing:** Xin Chang.

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
