## [Decision Letter · Decision Letter 0]

19 Apr 2022

PONE-D-21-34169A Determinate method of“ metrology attribute benchmark of Commercial Banks" Management EfficiencyPLOS ONE

Dear Dr. Chang,

Thank you for submitting your manuscript to PLOS ONE. After careful consideration, we feel that it has merit but does not fully meet PLOS ONE’s publication criteria as it currently stands. Therefore, we invite you to submit a revised version of the manuscript that addresses the points raised during the review process.

We look forward to receiving your revised manuscript.

Kind regards,

Hua Wang

Academic Editor

PLOS ONE

Journal Requirements:

Reviewers' comments:

Reviewer's Responses to Questions

**Comments to the Author**

1. Is the manuscript technically sound, and do the data support the conclusions?

Reviewer #1: Yes

Reviewer #2: Partly

2. Has the statistical analysis been performed appropriately and rigorously? 

Reviewer #1: Yes

Reviewer #2: No

3. Have the authors made all data underlying the findings in their manuscript fully available?

Reviewer #1: Yes

Reviewer #2: Yes

4. Is the manuscript presented in an intelligible fashion and written in standard English?

Reviewer #1: Yes

Reviewer #2: Yes

5. Review Comments to the Author

Reviewer #1: Review Report

Name of the Journal: PLOS ONE

Manuscript id: PONE-D-21-34169

Title of the manuscript: A Determinate method of“ metrology attribute benchmark of Commercial Banks" Management Efficiency

The study utilizes cluster mining technology to analyze the attribute data of several commercial banks based on mastering the management efficiency metrology attribute data, mining their characteristics and patterns, and determines the candidate benchmark set. In the candidate benchmark, the metrology attribute benchmark is determined by using average value, thus obtaining a universal metrology attribute benchmark. The study employs the statistical clustering analysis technique in the field of efficiency management .

Comments and Suggestions.

• Replace “ belongs” by “belong “ in the statement: “Let the condensed point set belongs to E” .y

• References are not uniformly written following the journal standard format. Check all the references.

Reviewer #2: 1. The authors started the abstract with the aim of the study as follows: (In order to address the inadequacies of current management efficiency evaluation in commercial banks, the idea of management efficiency metrology is proposed.). In my opinion, they should start with the context and background of the topic.

2. again, reorganize the abstract to conclude:

(a) The overall purpose of the study and the research problems you investigated.

(b) The basic design of the study.

(c) Major findings or trends found as a result of the study.

(d) A brief summary of your interpretations and conclusions.

3. The overall presentation of the manuscript is confusing to the readers. The authors should summarise the study and presented it again with a simplified sequence of ideas without complexity.

4. The results and discussion for the presented study need to be improved.

6. PLOS authors have the option to publish the peer review history of their article (what does this mean?). If published, this will include your full peer review and any attached files.

Reviewer #1: **Yes: **Dr. Surapati Pramanik

Reviewer #2: No

---

## [Author Response · Author response to Decision Letter 0]

21 Jun 2022

Reviewer#1, Concern # 1: Replace “belongs” by “belong” in the statement: “Let the condensed point set belongs to E”. 

Author response: We updated the manuscript by replacing “belongs” by “belong” in the statement: “Let the condensed point set belongs to E”.

Author action: We updated the manuscript by replacing “belongs” by “belong” in the statement: “Let the condensed point set belongs to E”. (Page7)

Reviewer#1, Concern # 2: References are not uniformly written following the journal standard format. Check all the references.

Author response: We have changed the reference format according to the journal requirements standard.

Author action: We updated the manuscript by changing the reference format according to the journal requirements standard. (Page16-17)

Reviewer#2, Concern # 1: The authors started the abstract with the aim of the study as follows: (In order to address the inadequacies of current management efficiency evaluation in commercial banks, the idea of management efficiency metrology is proposed.). In my opinion, they should start with the context and background of the topic. 

Author response: We updated the manuscript by rewriting the abstract and adding a summary of background significance.

Author action: We updated the manuscript by rewriting the abstract and adding a summary of background significance. (Page1)

Reviewer#2, Concern # 2: again, reorganize the abstract to conclude:

(a) The overall purpose of the study and the research problems you investigated.

(b) The basic design of the study.

(c) Major findings or trends found as a result of the study.

(d) A brief summary of your interpretations and conclusions. 

Author response: We updated the manuscript by rewriting

(a) The overall purpose of the study and the research problems you investigated.

(b) The basic design of the study.

(c) Major findings or trends found as a result of the study.

(d) A brief summary of interpretations and conclusions.

Author action: We updated the manuscript by revising the abstract and reorganizing the abstract from four aspects proposed by expert. (Page1)

Reviewer#2, Concern # 3: The overall presentation of the manuscript is confusing to the readers. The authors should summarize the study and presented it again with a simplified sequence of ideas without complexity.

Author response: We have read the full text and revised the introduction and further emphasized the research ideas in the introduction.

Author action: We updated the manuscript by revising introduction (page3, Sec.1), describing and addressing our highlights and novel points in introduction.

Reviewer#2, Concern # 4: The results and discussion for the presented study need to be improved.

Author response: Aim to this concern, we have revised the Conclusions to make the conclusion more concise and to highlight the main work of this paper.

Author action: We updated the manuscript by rewriting the conclusion. (Page15, Sec.4)

---

## [Decision Letter · Decision Letter 1]

18 Jul 2022

A Determinate Method of Metrology Attribute Benchmark of Commercial Banks'  Management Efficiency

PONE-D-21-34169R1

Dear Dr. Chang,

We’re pleased to inform you that your manuscript has been judged scientifically suitable for publication and will be formally accepted for publication once it meets all outstanding technical requirements.

Kind regards,

Hua Wang

Academic Editor

PLOS ONE

Additional Editor Comments (optional):

Reviewers' comments:

Reviewer's Responses to Questions

**Comments to the Author**

1. If the authors have adequately addressed your comments raised in a previous round of review and you feel that this manuscript is now acceptable for publication, you may indicate that here to bypass the “Comments to the Author” section, enter your conflict of interest statement in the “Confidential to Editor” section, and submit your "Accept" recommendation.

Reviewer #1: All comments have been addressed

Reviewer #2: All comments have been addressed

2. Is the manuscript technically sound, and do the data support the conclusions?

Reviewer #1: Yes

Reviewer #2: Yes

3. Has the statistical analysis been performed appropriately and rigorously? 

Reviewer #1: Yes

Reviewer #2: Yes

4. Have the authors made all data underlying the findings in their manuscript fully available?

Reviewer #1: Yes

Reviewer #2: Yes

5. Is the manuscript presented in an intelligible fashion and written in standard English?

Reviewer #1: Yes

Reviewer #2: Yes

6. Review Comments to the Author

Reviewer #1: Review Report

Journal: PLOS ONE

Manuscript Number: PONE-D-21-34169R1

Title: A Determinate Method of Metrology Attribute Benchmark of Commercial Banks'

Management Efficiency

The authors have addressed all the comments and suggestions and the manuscript has been improved.

Reviewer #2: Dear authors

All comments have been addressed in the revised version.

Congratulations and best wishes

Regards

7. PLOS authors have the option to publish the peer review history of their article (what does this mean?). If published, this will include your full peer review and any attached files.

Reviewer #1: **Yes: **Dr. Surapati Pramanik

Reviewer #2: No

---

## [Editor Report · Acceptance letter]

26 Jul 2022

PONE-D-21-34169R1 

A Determinate Method of Metrology Attribute Benchmark of Commercial Banks' Management Efficiency 

Dear Dr. Chang:

I'm pleased to inform you that your manuscript has been deemed suitable for publication in PLOS ONE. Congratulations! Your manuscript is now with our production department. 

Kind regards, 

on behalf of

Dr. Hua Wang 

Academic Editor

PLOS ONE